# Convolution of Barker and Golay Codes for Low Voltage Ultrasonic Testing

**Zeng Fan [1], John Rudlin [2], Giorgos Asfis [2] and Hongying Meng [1,\*]**

[1] Department of Electronic and Computer Engineering, Brunel University London, Uxbridge UB8 3PH, UK; Zeng.Fan@brunel.ac.uk

[2] TWI Ltd. Granta Park, Cambridge CB21 6AL, UK; john.rudlin@twi.co.uk (J.R.); Giorgos.Asfis@twi.co.uk (G.A.)

\* Correspondence: hongying.meng@brunel.ac.uk; Tel.: +44-(0)1895265496

**Abstract:** Ultrasonic Testing (UT) is one of the most important technologies in Non-Detective Testing (NDT) methods. Recently, Barker code and Golay code pairs as coded excitation signals have been applied in ultrasound imaging system with improved quality. However, the signal-to-noise ratio (SNR) of existing UT system based on Barker code or Golay code can be influenced under high high attenuation materials or noisy conditions. In this paper, we apply the convolution of Barker and Golay codes as coded excitation signals for low voltage UT devices that combines the advantages of Barker code and Golay code together. There is no need to change the hardware of UT system in this method. The proposed method has been analyzed theoretically and then in extensive simulations. The experimental results demonstrated that the main lobe level of the code produced by convolution of Barker code and Golay code pairs is much higher than the simple pulse and the main lobe of the combined code is higher than the traditional Barker code, sidelobe is the same as the baker code that constitutes this combined code. So the peak sidelobe level (PSL) of the combined code is lower than the traditional Barker code. Equipped with this, UT devices can be applied in low voltage situations.

**Keywords:** ultrasonic testing; coded excitation; SNR; Golay code; Barker code

## 1. Introduction

Ultrasonic Testing (UT) is an important technology in non-destructive testing (NDT) methods that is often used for obtaining parameters related to the durability, the physical and mechanical properties of material by using ultrasound wave. The traditional pulse-echo and transmission techniques were applied [1] for detecting flaws in the material. In the case of certain hazardous flammable and explosive atmospheres, traditional high-voltage UT equipment cannot be used. The main reason UT equipment is unsuitable is the high voltage used by the pulsing circuitry. It has both enough energy to ignite a gas but also has the potential to create a spark in case of a failure. However, the lower the pulsing voltage is, the lower the signal-to-noise ratio (SNR) of the signal is. Attempting to reduce the pulsing voltage can lead to noisy and unusable results. In order to ensure high SNR of signals in low-voltage ultrasonic testing equipment, a method without changing the hardware of the UT device should be considered. The technique of coded excitation has been used to improve the SNR in ultrasonic testing [2]. The technique of coded excitation has been used in radar for about 60 years but it was successfully introduced for clinical ultrasound only within the last 10 years [3]. Coded excitation refers to the technique of using a series of pulses following a calculated width and pause structure. These pulse trains follow a mathematically described code sequence and are used as complementary pairs whose auto-correlation coefficients sum to zero in most cases. The most well-known and studied pairs of coded sequences are the Golay code and Barker code [4]. The result of such an excitation is a long sequence and it must be associated with a transmitted signal, so that the received echo

becomes shorter in duration and higher in intensity, thereby increasing the system SNR. By using the compressed signal, clearer A-scan can be reconstructed. The long duration of the pulse trains allows for a significant amount of ultrasonic energy to be transmitted in the material. Thus, this technique is quite suitable for testing of highly attenuation materials using low-voltage pulse signal since the maximum amount of energy is not limited by the amplitude of the pulse [5].

In this paper, a coded excitation method which uses convolution of a Barker code and Golay code pair is applied for a low voltage ultrasonic testing system. The method combines the advantages of these two coding methods and increases the diversity of code lengths. The method can be used in a low-voltage ultrasonic testing device without changing the internal hardware. The main contributions of this paper are presented below.

- We investigated the combined coded excitation technique for a low-voltage ultrasonic testing device, not like other existing works which focus on high-voltage situations.
- We provided extensive theoretical analysis and simulation results for all the possible combination of coded excitation for ultrasonic testing, not like other works on selected coding methods.
- Current combined coded excitation techniques are applied in ultrasonic imaging system only. This work opens the door for all ultrasonic testing devices for many other applications in a low-voltage situation.

The rest of this paper is organized as follows. In Section 2, we review the related work of ultrasonic testing and coded excitation. In Section 3, we propose our methodology and analyze its superiority. The theoretical analysis results are described in Section 4 and the simulation results are described in Section 5. Finally, the conclusion and discussion are given in Section 6.

## 2. Related Work

Coded excitation applied in medical ultrasound imaging system was firstly proposed by Takeuchi [6] in 1979. However, due to the time-bandwidth limitation of this technique, there are rare studies on coded excitation in the literature in the following years. O'Donnell [7] considered to use coded excitation to improve the SNR of ultrasound system. A signal which has been compressed on the digital beamformer was used in the ultrasound system. Later, Jedwab and Parker [8] gave a general construction for an odd length binary Golay sequence pair of length 26 from a Barker sequence of length 13 and a related Barker sequence of length 11. This work lays the theoretical foundation for coded excitation.

Kim et al. [2] proposed a new coded excitation method by modulating Golay code pairs with Barker sequence for ultrasound imaging. However, they only used finite types of codes which are 3-bit Barker code, 8-bit Golay code pair and 3-bit Barker code modulated 4-bit Golay code pair in their experiment [2]. Recently, Wang and Cong [9] proposed a new pseudo Chirp-Barker-Golay (PCBG) coded excitation method in ultrasound imaging. They used the Field II toolbox to simulate phantom and cyst phantom of B-mode image.

Methodology-wise, there are two pulse compression methods. The first one is frequency modulated excitation signals, which include linear chirp code and pseudo-chirp excitation. The other one is binary coded excitation signals which include Barker code such as m-sequences, orthogonal Golay sequences. Coded excitation sequences operate in a slightly different way compared to frequency modulated excitation signals, the most common way for this technique is the burst cascade coding according to the polarity of binary sequence such as a sequence composed of 1 and 0 or $-1$ [10].

The limitation is the length of the sequences of the coded excitation. The improvement of SNR is related to the length of the sequences but there are only a few lengths of Barker codes available, although there is no limitation for conventional Golay codes [3]. Even if the length of selected codes is not the required length, one of the codes of a known length is inevitably selected. Therefore, this may lead to inefficiency.

So far, the combined code excitation methods are only applied in ultrasound imaging under high-voltage situations and the selection of the combined code is limited. However, for other ultrasound devices under a low-voltage situation, there are no existing studies. Here, we investigate this task from both complete theoretical analysis and extensive simulation experiments.

## 3. Materials and Methods

### 3.1. Matched Filter

Matched filters which are electrical tools for detecting a known piece of signal is widely used in radar, mobile communication systems and ultrasound imaging as an important filter [11]. The optimal criterion for the matched filter is that the output SNR is the largest when it is in the white noise background and noise bandwidth is wider than signal bandwidth. Since the transmitted signal spans a wide frequency range, the matched filter cuts out some of the signal as well as some background noise because it has a finite bandwidth [12]. The frequency response of the matched filter is the conjugate of the frequency response of the input signal. The basic problem that often arises in ultrasonic testing system is that detecting a pulse transmitted over a channel is corrupted by channel noise. The Equation (1) shows the ideal receiving structure of Barker code signal $B(t)$ though a matched filter.

$$
\begin{aligned}
y(t) &= x(t) * h(t) \\
x(t) &= B(t) + w(t) \\
h(t) &= B(T - t)
\end{aligned}
\tag{1}
$$

where $w(t)$ is the white noise and $x(t)$ is the filter input which is the sum of $B(t)$ and $w(t)$. $y(t)$ is the filter output which is the convolution of filter input $x(t)$ and matched filter $h(t)$ when T is the length of $B(t)$ [13].

From amplitude-frequency characteristics, amplitude-frequency characteristics of matched filter are related to the amplitude-frequency characteristics of input signal. At the stronger signal frequency point, the amplification factor of the filter is also larger. The matched filters allow received signal to pass as much as possible regardless of the characteristics of the noise. Since one of the preconditions of the matched filter is white noise, the power spectrum of the noise is flat. In this case, this technique can reduce the passage of noise.

From phase-frequency characteristics, phase-frequency characteristics of the matched filter are conjugated to the input signal. Thus, the phase of the signal is zero by matching the filter. This can achieve a coherent superposition of the signal in the time domain. The phase of the noise is random and only non-coherent superposition can be achieved. This technique ensures the maximum SNR of the output signal in the time domain. In fact, the name of the matched filter indicates its distinctive feature. The matched filters match the input signals. Once the input signal changes, the original matched filter also changes.

### 3.2. Barker Code

The Barker code is one of the binary phase codes that produce a compressed waveform with a constant sidelobe level equal to one. The auto-correlation coefficients are defined and should all be small [14]:

$$
\left| \sum_{i=1}^{N-k} b_i b_{i+k} \right| \leq 1
\tag{2}
$$

In the Equation (2), $b_i$ is an element of Barker code $B_N$ and $b_i = \pm 1$. N is the length of the Barker code. $N \geq 2$ and $1 \leq k < N$. There are Barker codes of lengths 2, 3, 4, 5, 7, 11 and 13, and it is conjectured that there is no longer a Barker code [15]. The optimal binary sequence is that whose auto-correlation peak sidelobe is the minimum possible for a given code length. The advantage is that the self-correlation or matched filtering of these codes gives the main lobe peak of $N$ and the

minimum peak sidelobe of 1. Table 1 lists all known Barker codes and the side lobes level of these codes. Ideally, if there is a long length, these codes can be used for pulse compression radar. However, the longest known Barker code length is 13, so the pulse compression radar using these Barker codes will be limited to the maximum compression ratio of 13. Figure 1 shows a 5-bit Barker code and the compression result after the matched filter. It can be seen that the frequency characteristics of the matched filter is exactly the same as the input signal and there are very few changes in the frequency characteristics of the compression signal.

Barker code has a good performance of inhibiting the auto-correlation side lobe. Due to the property of the Barker code which can improve the SNR, it is considered as the transmission code for the low-voltage ultrasonic device.

**Table 1.** Barker code of various lengths.

| CODE SYMBOL | LENGTH | CODE | SIDE-LOBE LEVEL (dB) |
|:---:|:---:|:---:|:---:|
| B2 | 2 | (+1, −1), (+1,+1) | −6 |
| B3 | 3 | +1, +1, −1 | −9.5 |
| B4 | 4 | (+1, −1, +1, +1), (+1, −1, −1, −1) | −12 |
| B5 | 5 | +1, +1, +1, −1, +1 | −14 |
| B7 | 7 | +1, +1, +1, −1, −1, +1, −1 | −16.9 |
| B11 | 11 | +1, +1, +1, −1, −1, −1, +1, − 1, −1, +1, −1 | −20.8 |
| B13 | 13 | +1, +1, +1, +1, −1, −1, −1, +1, +1, −1, +1, −1, +1 | −22.3 |

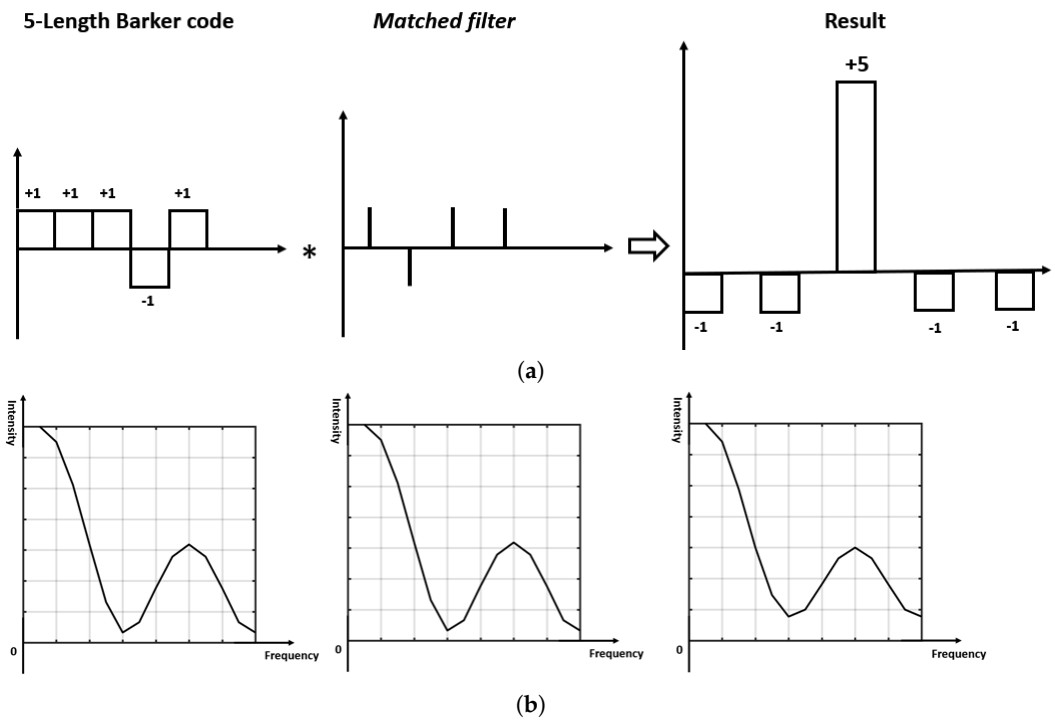

**Figure 1.** Illustration of 5-bit Barker code compression. (**a**) A pulse compression example of 5-bit Barker code compression. (**b**) Frequency response of the 5-bit Barker code, matched filter and compressed result.

### 3.3. Golay Code

Golay code is defined as a pair of equal finite length composed elements and the number of identical element pairs in one sequence is equal to the number of distinct element pairs in another sequence at any given interval [16]. An ideal pulse compression result without sidelobes is obtained by adding matched pairs of echoes which are generated by transmitting a pair of complementary sequences.

Defined is a pair of Golay complementary pair sequence $G_a(n)$ and sequence $G_b(n)$. Each element of $G_a(n)$ and $G_b(n)$ is either 1 or $-1$. Due to the property of Golay code pair, the auto-correlations of sequence $G_a(n)$ and sequence $G_b(n)$ are shown in Equation (3). $\delta(n)$ is a Dirac delta function. $*$ is the symbol of convolution.

$$G_a(n) * G_a(-n) + G_b(n) * G_b(-n) = 2N * \delta(n) \tag{3}$$

When use the coded excitation in ultrasonic testing or imaging, sequence $G_a(n)$ and sequence $G_b(n)$ are used instead of the traditional single-pulse excitation to stimulate the ultrasound transducer sequentially. The echoes from the same stationary object in the ultrasound field are represented by $e_a(n)$ and $e_b(n)$.

$$e_a(n) = y_t(n) * G_a(n) * y_r(n) \tag{4}$$

$$e_b(n) = y_t(n) * G_b(n) * y_r(n) \tag{5}$$

where $y_t(n)$ and $y_t(n)$ are the impulse response of the ultrasound transmitter and receiver. If the receiver and transmitter have the same element, $y_r(n) = y_t(n)$ the cross-correlation between $e_a(n)$ and sequence $G_a(n)$ is

$$e_a(n) * G_a(-n) = y_t(n) * G_a(n) * G_a(-n) * y_r(n) \tag{6}$$

And the cross-correlation between $e_b(n)$ and sequence $G_b(n)$ is

$$e_b(n) * G_b(-n) = y_t(n) * G_b(n) * G_b(-n) * y_r(n) \tag{7}$$

Sum the two cross-correlation results,

$$g(n) = e_a(n) * G_a(-n) + e_b(n) * G_b(-n) = 2N(y_t(n) * \delta(n) * y_r(n)) \tag{8}$$

$g(n)$ is the result of pulse compression of Golay code pair. It shows that the amplitude of $g(n)$ is equal to $2N$ times the echo amplitude of a single-pulse simulation. Figure 2 shows 4-bit Golay code pair and the compressed result after the matched filter.

Due to the unique properties, Golay code pairs can be used as the input signals of ultrasonic transmitter, the sidelobes of the echo signal after decoding have disappeared.

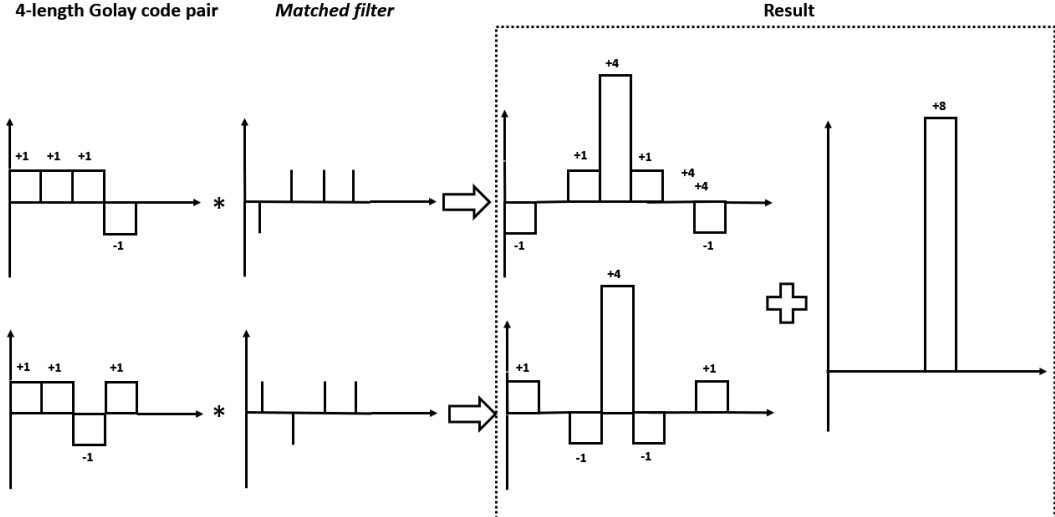

**Figure 2.** A pulse compression example of a 4-bit Golay code pair.

### 3.4. Convolution of Barker Code and Golay Code

Due to the character of the code excitation, the improvement of SNR and resolution is related to the code length. However, Golay codes and Barker codes are known as its specific sequence, it is only allowing to choose one of the sequences for ultrasonic testing. Barker-sequence combined Golay code (BCG) are convoluted by Golay code and Barker code, which can lead to additional improvement in signal intensity and flexibility for code length compared to convention Golay code pairs and Barker code. The combined codes have the same sidelobe of the baker code used and the main lobe of the combined codes is associated with the main lobe of Barker code and Golay code pairs. It combines the characteristics of a high main lobe Barker code and no side lobe Golay code pairs.

To excite the ultrasonic signal, the BCG code modulates the ultrasonic wave with a sine wave signal frequency carrier and then transmit beamformer and electrical transducer transmit the BCG code ultrasonic signal. The transducer can receive the echo signals from medium. Figure 3 shows the process of generation and compression of BCG codes. The BCG code pair is created by the convolution of Barker code and Golay code pair. The BCG code pairs are modulated and transmitted to the medium to obtain a pair of echoes. There are two steps to compress the echos. The first step is to process though Golay code pair matched filter and the second step is to process though Barker code matched filter. The compression result is the sum of the processed pairs.

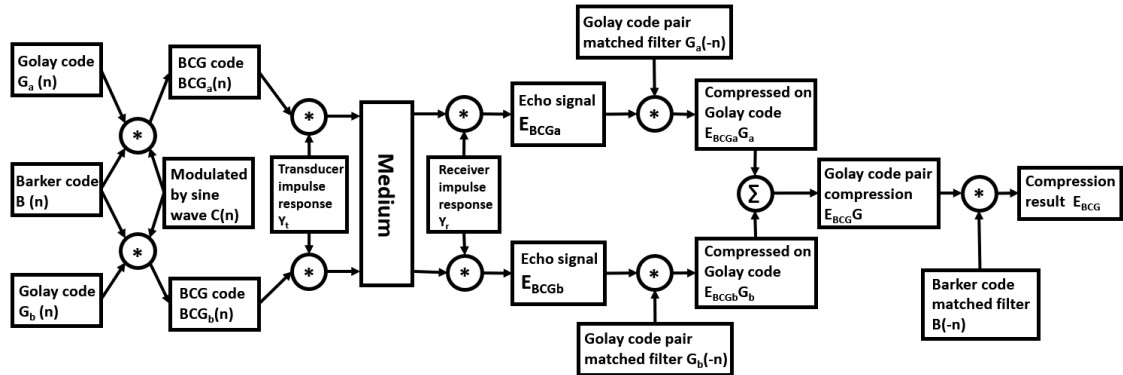

**Figure 3.** The process of generation and compression of Barker-sequence combined Golay codes (BCG).

A BCG code pair, $BCG_a$ and $BCG_b$, are built by using convolution of Golay sequence and Barker sequence and modulated by a base pulse as shown in following equations.

$$BCG_a(n) = G_a(n) * B(n) * C(n) \tag{9}$$

$$BCG_b(n) = G_b(n) * B(n) * C(n) \tag{10}$$

where $B(n)$ is a known Barker code sequence and $G_a(n)$, $G_b(n)$ are the Golay code pairs. Defined the $BCG_a(n)$ and $BCG_b(n)$ as modulated BCG code signal. $C(n)$ is a carrier sine wave. $y_t(n)$ and $y_t(n)$ are the impulse response of the ultrasound transmitter and receiver.

$$E_{BCGa}(n) = y_t(n) * BCG_a(n) * y_r(n) \tag{11}$$

$$E_{BCGb}(n) = y_t(n) * BCG_b(n) * y_r(n) \tag{12}$$

Then the $E_{BCGa}(n)$ and $E_{BCGb}(n)$ are the beamformed echo signal of $BCG_a(n)$ and $BCG_b(n)$. The received signal will be processed by matched filter. Due to the composite signal is combined by two types of signal. The received signal should be processed by the matched filter twice. The first step is to process the signal by using a corresponding Golay code pair matched filter, the second step is to

process the first step signal by using a corresponding Barker sequence matched filter. The echo signals which are processed by Golay matched filter can be defined as

$$E_{BCGa}G_a = E_{BCGa}(n) * G_a(-n) = y_t(n) * BCG_a(n) * y_r(n) * G_a(-n) \tag{13}$$

$$E_{BCGb}G_b = E_{BCGb}(n) * G_b(-n) = y_t(n) * BCG_b(n) * y_r(n) * G_b(-n) \tag{14}$$

where $E_{BCGa}G_a(n)$ and $E_{BCGb}G_b(n)$ are the echo signals processed by Golay matched filter and $G_a(-n)$ and $G_b(-n)$ are known as the matched filter of each Golay code pair. Sum the result of each compressed signal can achieve the Golay code compression of the BCG code as:

$$
\begin{aligned}
E_{BCG}G &= E_{BCGa}G_a + E_{BCGb}G_b \\
&= y_t(n) * y_r(n) * (G_a(-n) * BCG_a(n) + G_b(-n) * BCG_b(n)) \\
&= y_t(n) * y_r(n) * (G_a(-n) * G_a(n) * B(n) + G_b(-n) * G_b(n) * B(n)) \\
&= g(n) * B(n)
\end{aligned} \tag{15}
$$

$E_{BCG}G$ is the result of Golay code pair compression. As mentioned above, $g(n)$ is the result of pulse compression of Golay code pair. The Barker matched filter defined as $B(-n)$. Then the Golay compressed signal should be process by Barker matched filter as:

$$E_{BCG} = E_{BCG}G * B(-n) = p(n) * B(n) * B(-n)) = g(n) * E_B(n) \tag{16}$$

$E_B(n)$ is the result of Barker code compression. Therefore, $E_{BCG}$ is the result which is processed by the Golay matched filter and the Barker matched filter. By using this method, the length of the BCG code can be expressed as

$$N_{BCG} = \sum_{i=1}^{j} N_i - j + 1 \tag{17}$$

where $j$ is the number of the codes used for modulation and $N_i$ is the length of each code.

## 4. Theoretical Analysis

To prove the feasibility of the BCG code, the theoretical analysis has been done. The 15 different types of BCG code are used in the analysis. Three different types of pulses have been used: a simple pulse, 7-bit Barker code and 4-bit Golay code pairs. Peak sidelobe level (PSL) is a major parameter that describes a code's properties [17] and it can be calculated based on the peak ($P_{peak}$) and mean ($P_{mean}$) power of lobe pulse as shown in Equation (18).

$$PSL = 20log(\frac{P_{peak}}{P_{mean}}) \tag{18}$$

The different lengths of the BCG codes are shown in Table 2. Figure 4a shows the comparison of different theoretical intensities of 15 types of BCG codes. Figure 4b shows that a 3-bit Barker code modulated 4-bit Golay code compared to simple pulse, 7-bit Barker code and 4-bit Golay code. The maximum enhanced SNR of the 15 types of BCG code is 46.36 dB compared to the simple pulse which is B13G8. The enhanced SNR of B3G4 is 27.6 dB compared to the simple pulse, the enhanced SNR of 7-bit Barker code is 16.9 dB compared to the simple pulse and the enhanced SNR of 4-bit Golay code pairs is 18.1 dB compared to the simple pulse.

**Table 2.** Different lengths and peak-sidelobe ratio of BCG code.

| CODE SYMBOL | | CODE LENGTH | PSL (dB) |
|---|---|---|---|
| Barker 3 | Golay 2 | 4 | −19 |
| | Golay 4 | 6 | −23.4 |
| | Golay 8 | 10 | −28.6 |
| Barker 5 | Golay 2 | 6 | −22 |
| | Golay 4 | 8 | −24.8 |
| | Golay 8 | 12 | −28.8 |
| Barker 7 | Golay 2 | 8 | −24.2 |
| | Golay 4 | 10 | −26.4 |
| | Golay 8 | 14 | −29.6 |
| Barker 11 | Golay 2 | 12 | −27.6 |
| | Golay 4 | 14 | −29.2 |
| | Golay 8 | 18 | −31.4 |
| Barker 13 | Golay 2 | 14 | −29.0 |
| | Golay 4 | 16 | −30.2 |
| | Golay 8 | 20 | −32.2 |

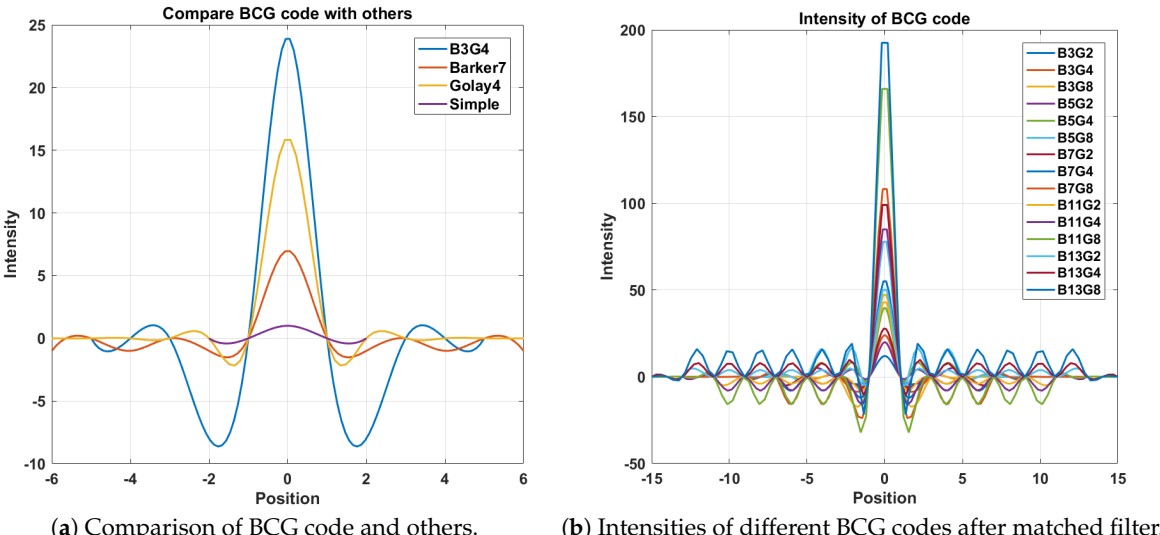

(**a**) Comparison of BCG code and others.     (**b**) Intensities of different BCG codes after matched filter.

**Figure 4.** The theoretical simulation results. (**a**) The 3-bit Barker code combined 4-bit Golay code compared to simple pulse, 7-bit Barker code and 4-bit Golay code. (**b**) the theoretical intensity comparison of 15 different types of BCG code.

## 5. Simulation

The theoretical results show that the BCG codes have a better performance than the simple pulse, Barker code and Golay code pairs in SNR and have a lower PSL than the Barker code. In order to verify the feasibility of this method under actual conditions, the ultrasonic testing simulation needs to be completed. Field II is a program based on linear acoustic in MATLAB. It can be used for simulating the production of the ultrasonic probe and the signal received by the probe to receive the sound field. This program can simulate the pulse generation and recall of various ultrasonic transducers in pulsed or continuous wave operation. In the simulation, the different types of pulse signals have been used as propagation signal though tissue.In order to get the echo signal, a scatterer has been set.

The frequency of the signal is 5 MHz and the sample frequency is 100 MHz. The speed of sound is a constant propagation speed of 1540 m/s. There are 32 elements used in the simulation. The width in the x-direction of the elements is 1 mm and the width in the y-direction of the elements is 5 mm; the kerf between each two element is 0.25 mm. The input signals have been modulated by carrier sine waves. The impulse responses of the pulse and receive apertures are a 2-cycle sine wave with a fit Hanning window. The voltage of the simple pulse is 1 V and input voltages of other codes are similar with a simple pulse. The purpose is for calculating the received signal from a collection of scatterers and for each combination of transmitting and receiving elements in the aperture. This corresponds to a full synthetic aperture scan, with each element transmitting and all elements receiving. This simulation experiment set a scatterer at [0, 0, 100] mm. Figure 5 shows the setup of simulation. There is an example of transmitted B3G4 code pair in Figure 6.

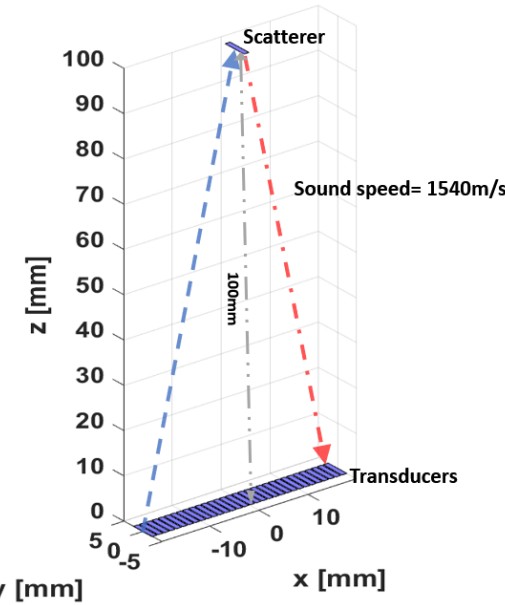

**Figure 5.** The simulation setup.

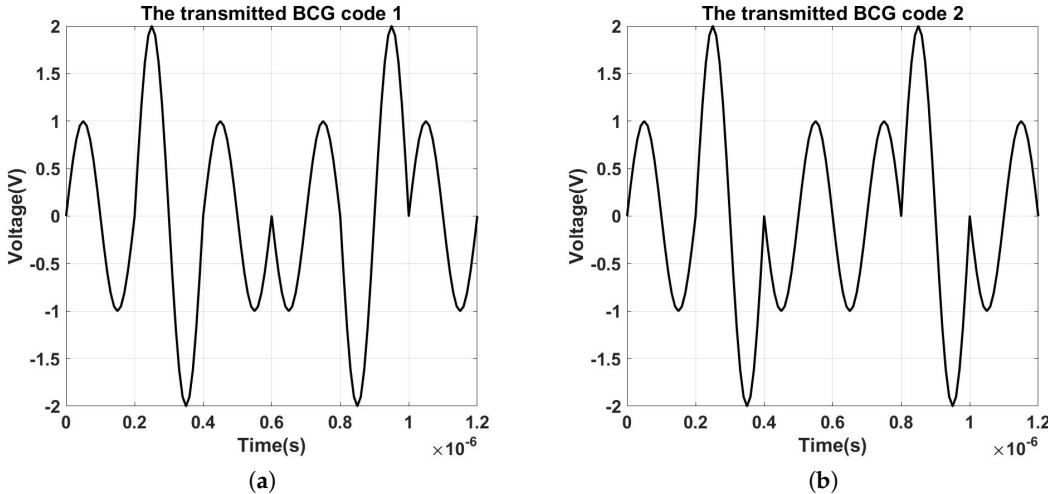

**Figure 6.** The transmitted B3G4 code pair in the Filed II simulation. (**a**) The first transmitted BCG code (**b**) The second transmitted BCG code.

Figure 7 shows the echo signal of simple pulse. The top figure shows composite image of received intensities from the individual elements of a linear array transducer. The bottom figure shows the central element of individual response.

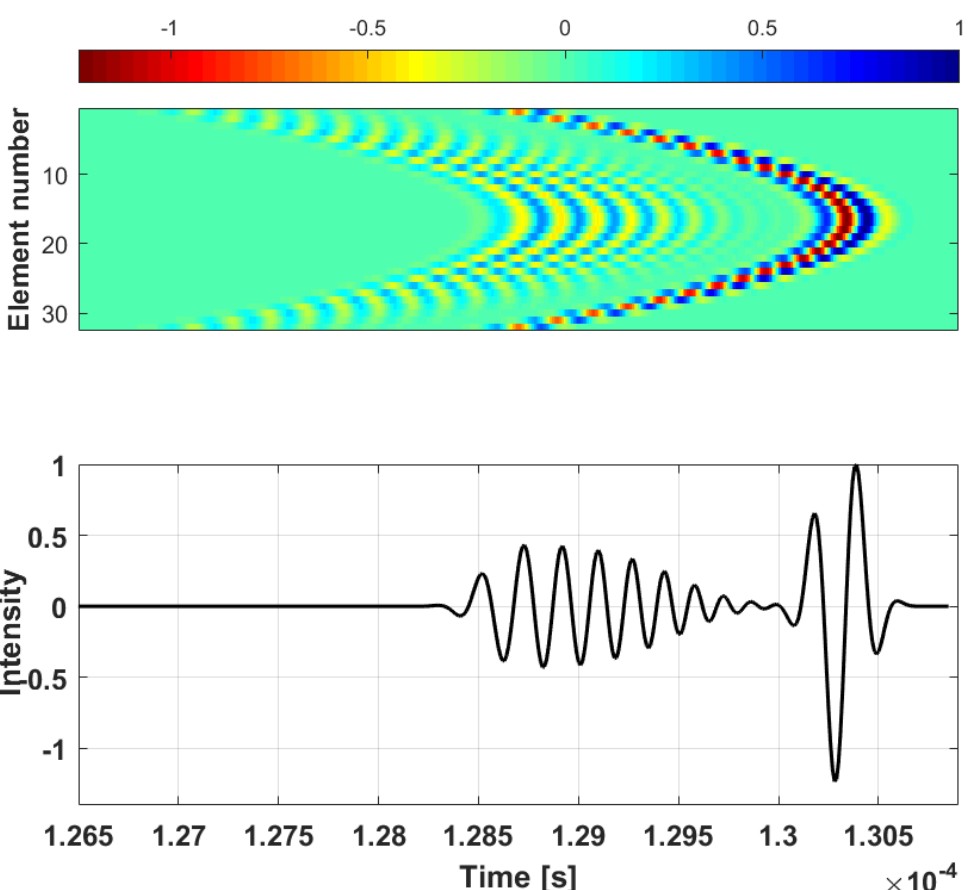

**Figure 7.** Received intensities from the individual elements of a linear array transducer (**top**) and the individual response of center element (**bottom**).

### 5.1. Barker Code and Golay Code Simulation Result

Figure 8 shows the received signal by using 7-bit Barker code signal. Figure 8a shows the original received signal, the top shows the composite image of received 7-bit Barker code intensities from the individual elements of a linear array transducer and the bottom shows the individual response of central element. Figure 8b shows the processed result of received signal after matched filter. The 7-bit Barker code had increased intensity of 18.5 dB compared to the simple Pulse. In the theoretical study, the increased intensity should be 16.9 dB.

Figure 9 shows the original echo results and compressed result of 4-bit Golay code pair signals. Figure 9a,b shows the first and second original echo signals, the top shows the composite image of received 4-bit Golay code intensities from the individual elements of a linear array transducer and the bottom shows the individual response of the central element. Figure 9c shows the processed result of received signal after matched filter and compression. The 4-bit Golay code pair had increased the intensity of 21.5 dB compared to the simple pulse.

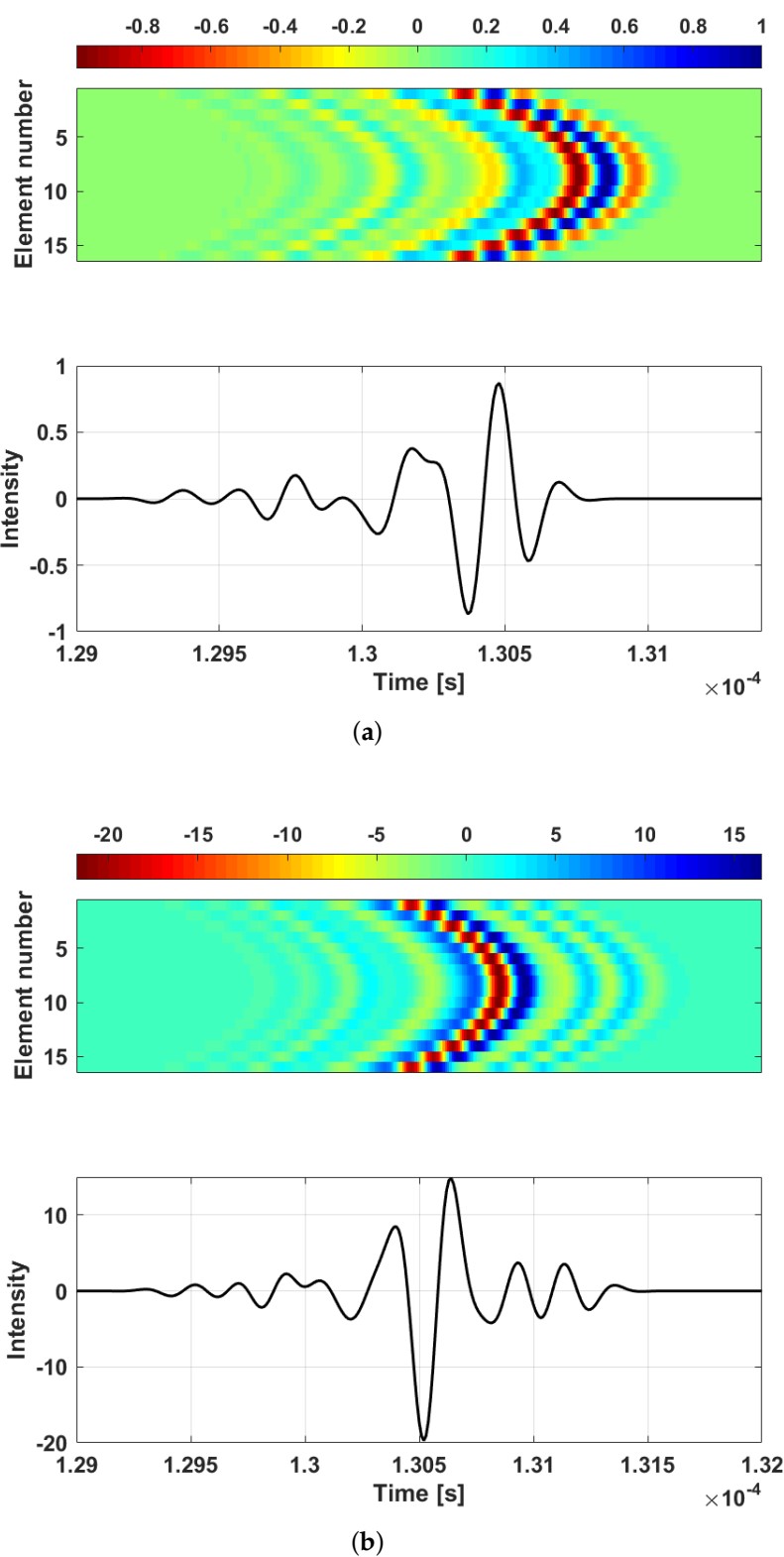

**Figure 8.** The 7-length Barker code simulation result. (**a**) The original received intensities from the individual elements of a linear array transducer (top) and the individual response of center element (bottom). (**b**) The compressed signals from the individual elements of a linear array transducer (top) and the compressed signal of center element (bottom).

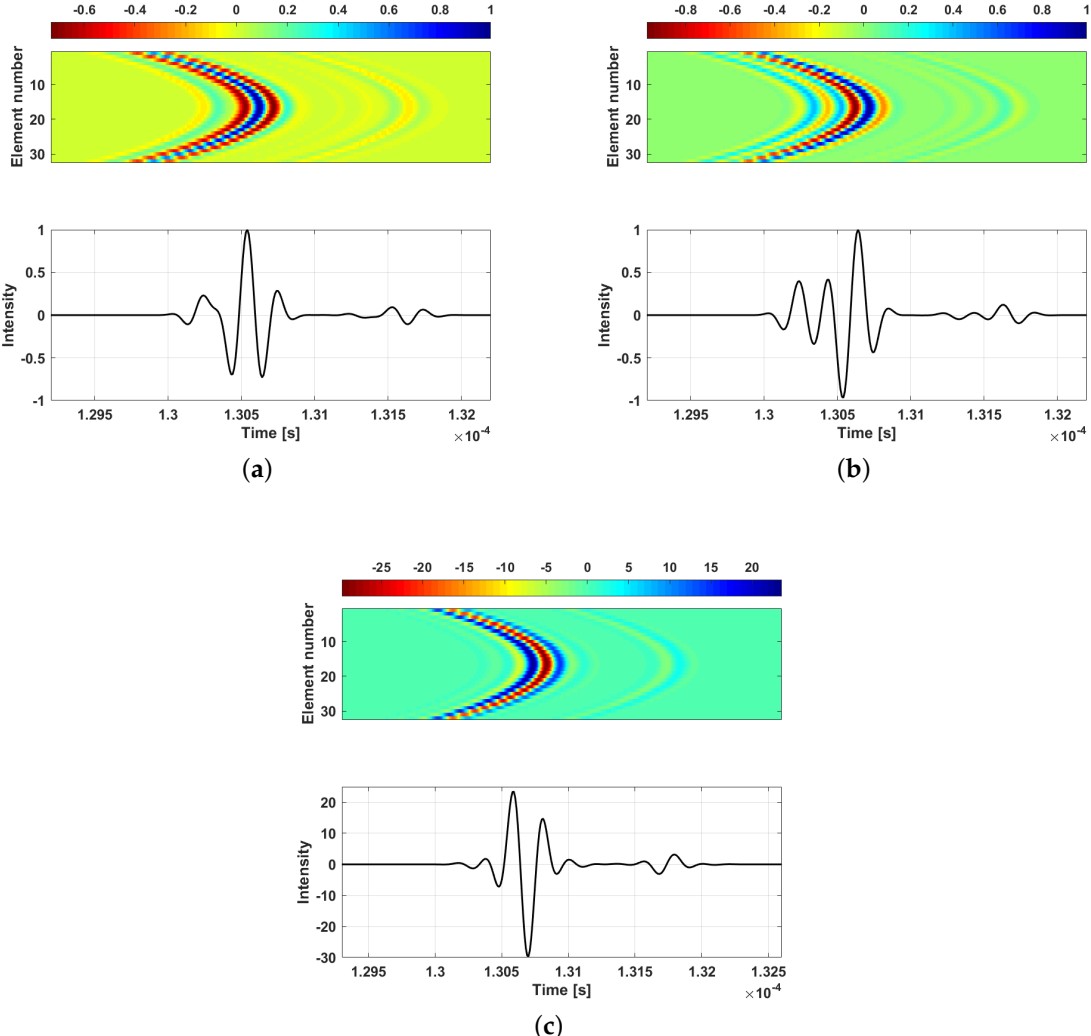

**Figure 9.** The 4-bit Golay code pair simulation results. (**a**) The original received 4-bit Golay code 1 intensities from the individual elements of a linear array transducer (top) and the individual response of center element (bottom). (**b**) The original received 4-bit Golay code 2 intensities from the individual elements of a linear array transducer (top) and the individual response of center element (bottom). (**c**) The compressed signals from the individual elements of a linear array transducer (top) and the compressed signal of center element (bottom).

### 5.2. Convolution of 3-Bit Barker Code and Various Length Golay Code

Figure 10 shows the results of convolution of 3-bit Barker code and various length Golay code which contain 3-bit Barker combine 2-bit Golay code pair, 3-bit Barker combine 4-bit Golay code pair and 3-bit Barker combine 8-bit Golay code pair. Each figure contains the original received signal part and the processed signal part. In both the original received signal part and the processed signal part, the composite images and central element of individual responses have been posted. By comparing these two part results, the increased intensities can be obviously seen. The B3G2 code had an increased intensity of 22.02 dB compared to the simple pulse and the signal side-lobe level is −18.9 dB. The B3G4 code had an increased intensity of 27.1 dB compared to the simple pulse and the signal side-lobe level is −19.2 dB. The B3G8 code had an increased intensity of 30.1 dB compared to the simple pulse and the signal side-lobe level is −23.0 dB.

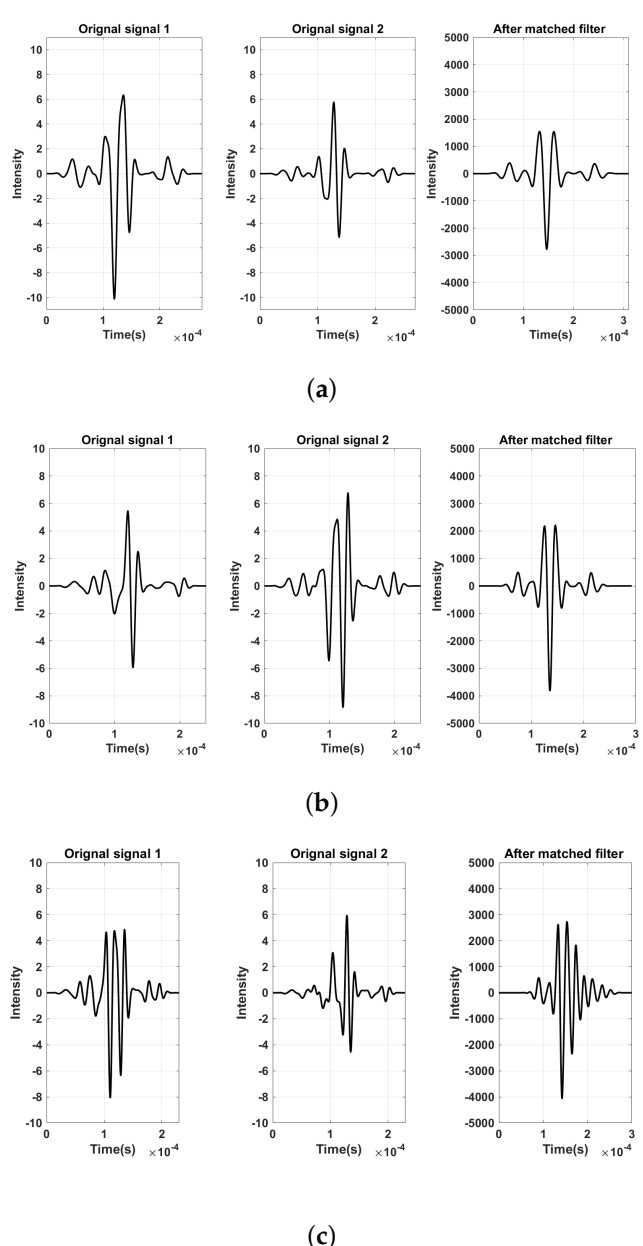

**Figure 10.** The central line of the 3-bit Barker combined different length Golay code pairs' simulation results. (**a**) The original and processed B3G2 pair signals. (**b**) The original and processed B3G4 signals. (**c**) The original and processed B3G8 pair signals.

*5.3. Convolution of 2-Bit Golay Code and Various Length Barker Code*

Figure 11 shows the results of the convolution of 2-bit Golay code and various lengths of Barker code which contain 5-bit Barker combine 2-bit Golay code pair, 7-bit Barker combine 2-bit Golay code pair, 11-bit Barker combine 2-bit Golay code pair and 13-bit Barker combine 2-bit Golay code pair. Each figure contains the original received signal part and the processed signal part. In both the original received signal part and the processed signal part, the composite images and central element of individual responses have been posted. By comparing these two part results, the increased intensities can be obviously seen. The B5G2 code had an increased intensity of 24 dB compared to the simple pulse and the PSL is −22.6 dB. The B7G2 code had an increased intensity of 27.78 dB compared to the simple pulse and the PSL is −24.2 dB. The B11G2 code had an increased intensity of 31.3 dB compared to the simple pulse and the PSL is −27.4 dB. The B13G2 code had an increased intensity of 35.1 dB compared to the simple pulse and the PSL is −28.2 dB.

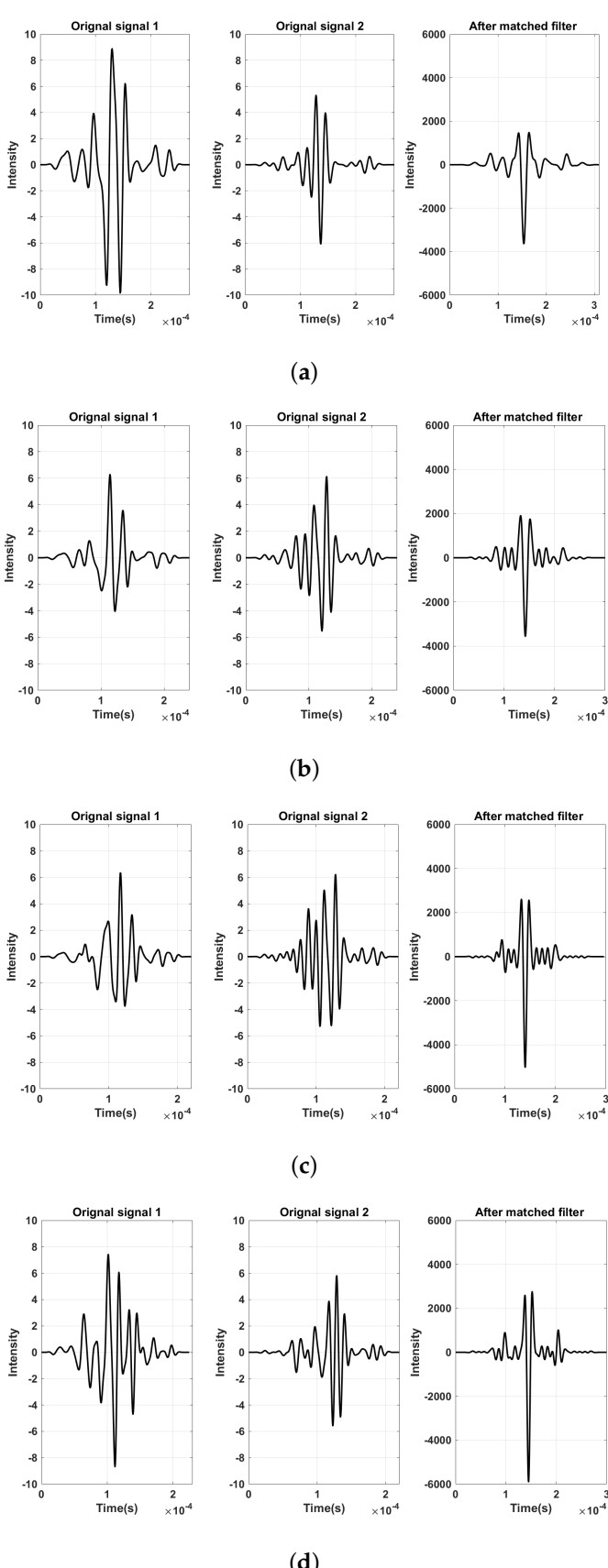

**Figure 11.** The central line of the 2-bit Golay code combine different length Barker code pairs' simulation results. (**a**) The original and processed B5G2 pair signals. (**b**) The original and processed B7G4 signals. (**c**) The original and processed B11G8 pair signals. (**d**) The original and processed B13G2 pair signals.

Table 3 shows all the SNR and PSL results from theory and the Matlab simulation. In Pilsu Kim's paper, the SNR of the 3-bit Barker code modulated 4-bit code exhibits the improvement of 27.68 dB in SNR and −18.66 dB in the sidelobe level [2] the results of which agree with the simulation result. One of the 15-bit M-sequences in the simulation module had an increased intensity of 17.8 dB compared to the simple pulse and the signal side-lobe level of −8.66 dB. One of the 15-bit Gold sequences in the simulation module had an increased intensity of 17 dB and the signal side-lobe level is −7.1 dB.

**Table 3.** Signal–noise ratio (SNR) and peak sidelobe level (PSL) results.

| CODE SYMBOL | CODE LENGTH | THEORETICAL RESULT | | SIMULATION RESULT | |
|:---:|:---:|:---:|:---:|:---:|:---:|
| | | SNR (dB) | PSL (dB) | SNR (dB) | PSL (dB) |
| Barker 7 | 7 | 16.9 | −17.0 | 18.5 | −18.1 |
| Golay 4 | 4 | 18.1 | - | 21.5 | - |
| B3G2 | 4 | 21.6 | −19 | 22.02 | −18.9 |
| B3G4 | 6 | 27.6 | −23.4 | 27.1 | −19.2 |
| B3G8 | 10 | 33.6 | −28.6 | 30.1 | −23.0 |
| B5G2 | 6 | 26.0 | −22.0 | 24 | −22.6 |
| B7G2 | 8 | 28.9 | −24.2 | 27.78 | −24.2 |
| B11G2 | 12 | 32.9 | −27.6 | 31.3 | −27.4 |
| B13G2 | 14 | 34.3 | −29.0 | 35.1 | −28.2 |
| M-sequence | 15 | 18.1 | −8.6 | 17.8 | −8.66 |
| Gold code | 15 | 18.1 | −8.6 | 17 | −7.1 |

## 6. Conclusions and Future Work

In this paper, we apply a convolution of Barker and Golay codes as coded excitation signals for low voltage UT devices. The combined codes have the same sidelobe as the Baker code and its main lobe is associated with that of the Barker code and Golay code pairs used. It combines the characteristics of a high main lobe Barker code and no side lobe Golay code pairs. There is no need to change the hardware of UT system in this method. The proposed method has been analyzed theoretically and then carried out in extensive simulation experiments. The experimental results demonstrated that the intensity of the code produced by the convolution of the Barker code and the Golay is much higher than the simple pulse and the PSL of the code is lower than that of the traditional Barker code. Equipped with this, any UT devices can be applied in low voltage situations for many applications. For the future work, we will implement them in hardware for real testing in the NDT and see whether it keeps the performance or not. Further, portable ultrasound NDT device will be developed for low voltage applications.

**Author Contributions:** Conceptualization, Z.F.; Methodology, Z.F.; Software, Z.F.; Validation, Z.F., G.A. and J.R.; Formal analysis, Z.F.; Investigation, Z.F.; Resources, Z.F. and G.A.; Data curation, Z.F.; Writing–original draft preparation, Z.F.; Writing–review and editing, Z.F. and H.M.; Visualization, Z.F.; Supervision, H.M. and J.R.; Project administration, J.R. and G.A.

**Funding:** This research was funded by TWI Ltd. and Brunel University London.

**Acknowledgments:** This publication was made possible by the sponsorship and support of TWI Ltd. and Brunel University London. The work was enabled through, and undertaken at, the National Structural Integrity Research Centre (NSIRC), a postgraduate engineering facility for industry-led research into structural integrity established and managed by TWI through a network of both national and international Universities.

**Conflicts of Interest:** The authors declare no conflict of interest.

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
