# Peer review of "Convolution of Barker and Golay Codes for Low Voltage Ultrasonic Testing"

_technologies, doi:10.3390/technologies7040072_

Round 1

Reviewer 1 Report

Paper looks like technical report, rather than scientific paper.

Rieview of other works should go first, there are several papers that already proposed and investigated the idea, e.g. :

A Construction of Binary Golay Sequence Pairs from Odd-Length Barker Sequences

Pseudo Chirp-Barker-Golay coded excitation in ultrasound imaging

Re [2] - you state "However,they only use limitation codes for their experiment". What is this?

In order to avoid naive statements, do the extensive study of spread spectrum signals: why and how. Many statements now are unclear or naive e.g.:

"Coded excitation sequences operate in a bit different way", "-1. Under any condition, an approximation of the initial pulse of the signal can be obtained", "there are only a few available lengths of .. Golay codes" - why this is not true. ", matched filter and the input signal have the same
amplitude-frequency characteristics", not quite true. barker code - should be Capital - its name.

Are you aware that xcorr not always=matched filter. When?

It is the length of the signal that defines the peak amplitude after correlation. Therefore fair comparison should use same length orf the signal or this has to be normalised somehow.

I do not understand what is your contribution, taking into account what other authors did?

Fig.1 needs revision - this is not matched filter, study better. This is some kind of measurement system but why it is there?

Figure 4. generation and compression of BCG codes - needs revision Use capital letter at each sentence start, test material is the same, not two different.

Improve and revise mauscript: English quality is low. E.g.:

"The results ... is", "beamformer that has been compressed was used in", "The matched filters allow receive signal","the BCG code transmits the ultrasonic signal through" [who transmits?], "The first time is processed by", "the meantime.Comparing with the exist result from other paper, the simulation results are also agreed with the experiment results.From" needs space. "only expands the number of coded excitation, but also"

Reviewer 2 Report

The paper is well written, there are some typos but nothing very important to point out. The paper describes the use of pulse compression to increase the SNR in Ultrasonic Testing. This is a well-known method extensible used in Radar or Sonar applications. The advantage of using convolution between Barker and Golay sequences are not well grounded. The main idea is to increase the sequence length by convolving two different types of sequences, however there is no evidence if the resultant auto-correlation properties are good or not. Such resultant sequences metrics must be compared with Gold or Kasami (or other similar) sequences with similar length.

Round 2

Reviewer 1 Report

Little improvement since last version:

1. This work is reporting same idea as [2] and [8]. Authors did not state this. On a contrary, they state that novelty of their approach is: "we propose a coded excitation method which uses convolution of Barker code and Golay code pair", while this is exactly the same what was reported in [2] and [8].

2. Authors state that their contribution is that "We investigated the combined coded excitation technique for low-voltage ultrasonic testing device, not like other existing works for high-voltage situation". This was not done. Only Matlab simulation was carried out using Field II toolbox. No real experiments.

3. Authors state that "We provided extensive theoretical analysis and simulation". What is presented is not extensive analysis nor extensive simulation, just simple exercise. Code sequences have been analysed directly, without accounting transducer contribution. Codes (Barker, Kasami, Gold, Golay) have the energy concentration at low frequencies and are not suitable for direct excitation. Therefore Phase Shift Keying (PSK) is used to upconvert to transducer passband. This is done by using CW toneburst (chip... not to be mixed with chirp).

4. Authors state that "The proposed method can be used in low-voltage ultrasonic testing device without changing the internal hardware". This is not true: code produced by Barker and Golay convolution is quaternary so it is not directly suitable for binary PSK modulation (0/180deg). Either PSK has to be combined with AM or polyphase PSK has to be used. In both case analog excitation is required, which requires significant modification of excitation electronics and FPGA hardware. Especially assuming that array ultrasound is aimed, as Fig. 5 demonstrates.  Yes, low voltage is less challenging, but why such complicated (quaternary) codes should be used if other, already existing, but binary, longer sequences can be employed?

5. Language used does not match the tongue accustom in the field. Furthermore, multiple wrong statement would mislead the possible readers about fundamental issues discussed. E.g.:

i)"the sidelobe level (SLL) of the code is lower than the traditional Barker code". This should not be the case-no matter what you do with Barker [convolution-transmission-correlation w Golay-sum-correlation with Barker // or ... correlation with convolved code // or same length Barker only], sidelobe level will be the same as for Barker only. Please check your Matlab.

ii)"existing UT system based on Barker code or Golay code suffered from lower signal-to-noise ratio (SNR) and low resolution". No. Coded excitation on contrary increases the signal level so the SNR. Furthermore, codes have nothing to do with resolution. Resolution is defined by received signal bandwidth, and this is mainly limited by transducer and chip duration used in PSK. Maybe it is the contrast that you want to address, but this is related to sidelobe level which suppose to be the best in case of Golay-only excitation. In your case sidelobe level is defined by Barker length.

iii) "The optimal criterion for the matched filter is that the SNR of output is the largest when it is in the white noise background". Not quite right. Matched filter (and also optimal filter) output is max when noise bandwidth is wider than signal BW. Majority of compression (processing gain) in not thanks to matched magnitude response of the filter (optimal filter does this too), but because of phase response inversion (time reversal is only in matched filter), but this applies only for spread spectrum signals (PSK, chirp).

iv) "Since the transmitted signal spans a wide frequency range, the matched filter cuts out some of the signal...". No. Matched filter passes all the components that REFERENCE signal contains.

v) Table 1 has wrong SLL's.

6. Some statements are trivial, but might lead to wrong assumptions. E.g.:

i)"The experimental results demonstrated that the intensity of the convolution of Barker code and Golay is much higher than the simple pulse" : first, the whole sentence has to be rewritten to make the statement clear [not the convolution but the code produced by convolution; intensity of convolution or mainlobe level?], then-this is trivial-code is longer so energy is higher, but this will be the case with every spread spectrum signal or even toneburst of equivalent length.

ii) "...Golay pairs and Barker code ... The result of such an excitation is a rather long pulse [are you aware what is pulse?] train whose reflections look like noise [are you sure?] and must be associated with received signal [hmm...], so that the received echo becomes shorter in duration and higher in intensity, thereby increasing the system resolution and SNR [??]. By using the processed signal [??], the meaningful [??] A-scan can be reconstructed..."

iii) "Barker code which can improve the SNR, it is considered as the transmitter"

iv) "The theoretical results show that the BCG codes have better performance than the simple code [??], Barker code and Golay code in SNR and SLL."

7. Statement "The method combines the advantages of these two coding methods.." is misleading: if only Golay code pairs are used, there should be no sidelobes. Meanwhile codes proposed do have a sidelobes which are defined by Barker length which is short in this case so sidelobes should be high.

Note on Golay code = no sidelobes. In reality this is not the case because transducer filters out some useful components and sidelobes leak out.

8. Authors state that "this technique is .. suitable for testing [of] highly attenuation [attenuative] materials", which is correct. So where are the experiments on such materials? Frequency dependent attenuation can be included in the simulation.

9. Eq (1) is wrong and has to be revised: (n)-what is this? Maybe (t)? If h(t) = B(-t) then x(t) is not B(t) + w(t), but  B(t-t0) + w(t) with B amplitude affected by attenuation, and if it is "highly attenuative material", then B(t) has to be altered in frequency domain to account the velocity and attenuation dispersion.  Statement "x(n) is the filter input which is the modulation of B(n) and w(n)" - wrong. Eq(2) is a complete mess - what it is about?

10.. No explanation has been presented how signal is transmitted. Read [2] and [8] to familiarise yourselves why this is needed. Eq (4)-(16) do not contain PSK or any other modulation function - keep in mind that that your codes are quaternary. Without this modulation signal can not be transmitted. By what can be deduced from manuscript, it seems that you tried direct excitation which will introduce significant losses since majority of energy is concentrated at 0 frequency.

11. Excersise with pure codes is trivial. Yet, if you do that, it must be done bitwise. I could not imagine how Fig.1 spectrum was obtained and what meaning does it make: read more literature on how coded sequences are used in ultrasound, particularly PSK. Code length should be stated not like "5-length Barker code" but "Barker codes of length 5" or simply "5-bit Barker code".

12. Consult what is SNR: now you are using it in a wrong way. Most probably it is a correlation peak height. In order to talk aboutSNR you would need noise. It could be SNR improvement, but first familiarise yourselves what is what and clearly state everything in manuscript.

13. Check the calculation of the SLL: results look wrong. Consult what theoretical values should be and use them properly. In order to compare the SLL, normalise the filters output for all signals to 1. In order to compare the compression performance, normalise to signal energy.

14. Consult what is optimal filter, matched filter, why processing gain (pulse compression) is occurring, what is signal base. There is no difference between matched filter and cross-correlation function if reference signal is noise-free.

15. Hanning window has no causality, its better to use simple filter.

16. English is still poor and needs significant revisions. E.g. "testing highly attenuation materials", "frequency response of the matched filter is the conjugate of the frequency of the input signal", "that no longer Barker codes exist","Golay code is defined as a pair of finite elements", "Sum the two cross-correlation", "Golay code can be used in ultrasonic transmitter", "side lobes
after decoding at the echo side are disappeared", "convention Golay code", "After comparing each performance", "The different lengths of BCG code are expressed at table 2", "Fig. 4 (a) shows that the theoretical intensity comparison", "the SNR of 4-length Golay code pairs is 18.1dB compare with", "Golay code compare with simple pulse", "Filed 2 is a toolbox [no such toolbox]", "stamping [???] frequency is 100MHz" , "receive aperture are", "Procedure for calculating the received signal from a collection of scatters and for each combination of transmit and receive elements in the aperture", "central element of individual response", "the bottom shows the central element of individual response", "code pair signals and. [??]", "code had increased intensity of 22.02dB", what is "portable ultrasound NDT device ...for low voltage applications"?

There are many more typing, style, language mistakes-it is not my task to hunt them down.

Fig.5 is not a simulation model but setup. Fig.7-9 you forgot to process in correlation or maybe even summation is not done  - something is wrong with signals which are titled "compression result after matched filter", "The compressed Golay pair signal", check the annotations. Fig.3 is wrong - are you sure there are two separate transducer pairs? Are you sure compression cannot use BCG code and two passages (Golay, then Barker) are needed?

Quality of figure should be increased so they are readable.

Number and dimension should have nonbreaking space in between, not 27.1dB, but 27.1{nonbreaking space}dB

My advise: state clearly that you are using the codes which have already been proposed in [2] and [8]. Value of your research can be in that you analyse how propagation (including transducer frequency response-both amplitude and phase) is affecting the SLL and mainlobe height compared to noise level - relative noise margin. Comparison should be between same length signals or at least same code length. It should be clearly stated that your codes are more complex so require different modulation scheme or analog excitation - arbitrary waveform generator. Or devise some binary (rectangular) modulation scheme. Reduce the explanatory part on pulse compression, be brief where extensive explanation is not necessary. But read thoroughly papers and books on this subject first. Then either do real experiment or simulation, but include the modulation part. Present all details, explain clearly. Then compare the SLL. Compare SNR , but with transmitted signal energy accounted.

Author Response

Thank you very much for the comments. Please see the attached file for our point to point response. 

Reviewer 2 Report

The authors response to my comments are accepted.  Nevertheless, this is a well-known method extensible used in Radar or Sonar applications and therefore it presents a lack of novelty. The paper could be accepted in the present form but the interest to the reader will be very low.

Author Response

Many thanks! 

Round 3

Reviewer 1 Report

No significant changes since last submission. Last time my note was the same -"no imrovement".

Authors just make cosmetic modifications and the state "We have thoroughly revised the manuscript to address those comments and concerns". No, this was not done and my major concerns have NOT been addressed.

For example:

Ethics issue: In previous review I indicated that "This work is reporting same idea as [2] and [8]. Authors did not state this. On a contrary, they state that novelty of their approach is: "we propose a coded excitation method which uses convolution of Barker code and Golay code pair", while this is exactly the same what was reported in [2] and [8]." that is, [2]and[8] have to be indicated as authors of this idea. This was not done, though rebuttal states that this was done.

My concern was that Only Matlab simulation was carried out using Field II toolbox. No real experiments.

Authors reply: "In the Field II toolbox, we have set the input voltages". Nonsense. This leads to nowhere.

My note that Phase Shift Keying (PSK) is used to upconvert to transducer passband was ignored. Furthermore, in rebuttal authors state that "In our simulation, the signal modulation and impulse responses have been already applied"

I indicated that statement "The proposed method can be used in low-voltage ultrasonic testing device without changing the internal hardware" is wrong and misleading: code requires analog excitation, so equipment HAS to be changed and properly programmed. And is extremely bulky. Author's reply is even more naive:"Universal waveform generator can be used as the ultrasonic generating device." There is no such thing like "universal", most probably "arbitrary". But this menas that equipment is bulky and such generator has 50ohm output impedance, which is usually not the case for driving the ultrasonic transducer: lots of energy will be lost.

Other concern, "Language used does not match the tongue accustom in the field" is made even worse: not they mention "baker code" (line 258)